# Syntax-Directed Variational Autoencoder for Structured Data

**Hanjun Dai[*1], Yingtao Tian[*2], Bo Dai[1], Steven Skiena[2], Le Song[1, 3]**

[1] College of Computing, Georgia Institute of Technology
[2] Department of Computer Science, Stony Brook University
[3] Ant Financial
[1] {hanjundai, bodai}@gatech.edu, lsong@cc.gatech.edu
[2] {yittian, skiena}@cs.stonybrook.edu

## Abstract

Deep generative models have been enjoying success in modeling continuous data. However it remains challenging to capture the representations for discrete structures with formal grammars and semantics, *e.g.*, computer programs and molecular structures. How to generate both syntactically and semantically correct data still remains largely an open problem. Inspired by the theory of compiler where the syntax and semantics check is done via syntax-directed translation (SDT), we propose a novel syntax-directed variational autoencoder (SD-VAE) by introducing *stochastic lazy attributes*. This approach converts the offline SDT check into on-the-fly generated guidance for constraining the decoder. Comparing to the state-of-the-art methods, our approach enforces constraints on the output space so that the output will be not only *syntactically* valid, but also *semantically* reasonable. We evaluate the proposed model with applications in programming language and molecules, including reconstruction and program/molecule optimization. The results demonstrate the effectiveness in incorporating syntactic and semantic constraints in discrete generative models, which is significantly better than current state-of-the-art approaches.

## 1 Introduction

Recent advances in deep representation learning have resulted in powerful probabilistic generative models which have demonstrated their ability on modeling continuous data, *e.g.*, time series signals (Oord et al., 2016; Dai et al., 2017) and images (Radford et al., 2015; Karras et al., 2017). Despite the success in these domains, it is still challenging to correctly generate discrete structured data, such as graphs, molecules and computer programs. Since many of the structures have syntax and semantic formalisms, the generative models without explicit constraints often produces invalid ones.

Conceptually an approach in generative model for structured data can be divided in two parts, one being the formalization of the structure generation and the other one being a (usually deep) generative model producing parameters for stochastic process in that formalization. Often the hope is that with the help of training samples and capacity of deep models, the loss function will prefer the valid patterns and encourage the mass of the distribution of the generative model towards the desired region automatically.

Arguably the simplest structured data are sequences, whose generation with deep model has been well studied under the seq2seq (Sutskever et al., 2014) framework that models the generation of sequence as a series of token choices parameterized by recurrent neural networks (RNNs). Its widespread success has encourage several pioneer works that consider the conversion of more complex structure data into sequences and apply sequence models to the represented sequences. Gómez-Bombarelli et al. (2016) (CVAE) is a representative work of such paradigm for the chemical molecule generation, using the SMILES line notation (Weininger, 1988) for representing molecules.

---

*Both authors contributed equally to the paper.

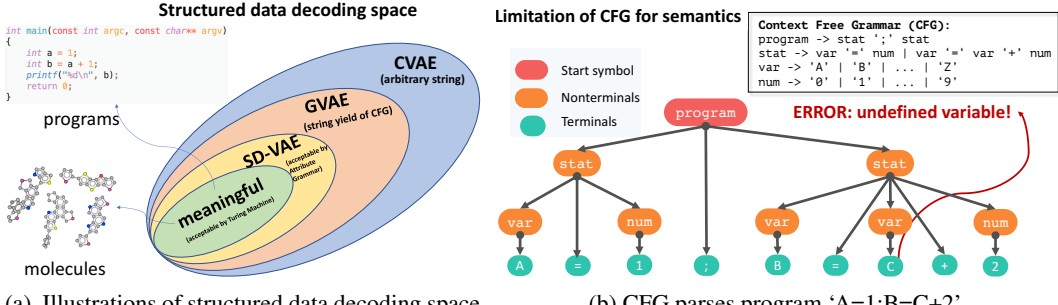

(a) Illustrations of structured data decoding space

(b) CFG parses program 'A=1;B=C+2'

Figure 1: Illustration on left shows the hierarchy of the structured data decoding space w.r.t different works and theoretical classification of corresponding strings from formal language theory. SD-VAE, our proposed model with attribute grammar reshapes the output space tighter to the meaningful target space than existing works. On the right we show a case where CFG is unable to capture the semantic constraints, since it successfully parses an invalid program.

However, because of the lack of formalization of syntax and semantics serving as the restriction of the *particular* structured data, underfitted *general-purpose* string generative models will often lead to invalid outputs. Therefore, to obtain a reasonable model via such training procedure, we need to prepare large amount of valid combinations of the structures, which is time consuming or even not practical in domains like drug discovery.

To tackle such a challenge, one approach is to incorporate the structure restrictions explicitly into the generative model. For the considerations of computational cost and model generality, context-free grammars (CFG) have been taken into account in the decoder parametrization. For instance, in molecule generation tasks, Kusner et al. (2017) proposes a grammar variational autoencoder (GVAE) in which the CFG of SMILES notation is incorporated into the decoder. The model generates the parse trees directly in a top-down direction, by repeatedly expanding any nonterminal with its production rules. Although the CFG provides a mechanism for generating *syntactic valid* objects, it is still incapable to regularize the model for generating *semantic valid* objects (Kusner et al., 2017). For example, in molecule generation, the semantic of the SMILES languages requires that the rings generated must be closed; in program generation, the referenced variable should be defined in advance and each variable can only be defined exactly once in each local context (illustrated in Fig 1b). All the examples require cross-serial like dependencies which are not enforceable by CFG, implying that more constraints beyond CFG are needed to achieve semantic valid production in VAE.

In the theory of compiler, attribute grammars, or syntax-directed definition has been proposed for attaching semantics to a parse tree generated by context-free grammar. Thus one straightforward but not practical application of attribute grammars is, after generating a syntactic valid molecule candidate, to conduct offline semantic checking. This process needs to be repeated until a semantically valid one is discovered, which is at best computationally inefficient and at worst infeasible, due to extremely low rate of passing checking. As a remedy, we propose the *syntax-direct variational autoencoder* (SD-VAE), in which a semantic restriction component is advanced to the stage of syntax tree generator. This allows the generator with both syntactic and semantic validation. The proposed syntax-direct generative mechanism in the decoder further constraints the output space to ensure the semantic correctness in the tree generation process. The relationships between our proposed model and previous models can be characterized in Figure 1a.

Our method brings theory of formal language into stochastic generative model. The contribution of our paper can be summarized as follows:

- *Syntax and semantics enforcement*: We propose a new formalization of semantics that systematically converts the offline semantic check into online guidance for stochastic generation using the proposed *stochastic lazy attribute*. This allows us effectively address both syntax and semantic constraints.

- *Efficient learning and inference*: Our approach has computational cost $O(n)$ where $n$ is the length of structured data. This is the same as existing methods like CVAE and GVAE which do not enforce semantics in generation. During inference, the SD-VAE runs with semantic guiding on-the-fly, while the existing alternatives generate many candidates for semantic checking.

- *Strong empirical performance*: We demonstrate the effectiveness of the SD-VAE through applications in two domains, namely (1) the subset of Python programs and (2) molecules. Our approach consistently and significantly improves the results in evaluations including generation, reconstruction and optimization.

## 2 BACKGROUND

Before introducing our model and the learning algorithm, we first provide some background knowledge which is important for understanding the proposed method.

### 2.1 VARIATIONAL AUTOENCODER

The variational autoencoder (Kingma & Welling, 2013; Rezende et al., 2014) provides a framework for learning the probabilistic generative model as well as its posterior, respectively known as decoder and encoder. We denote the observation as $x$, which is the structured data in our case, and the latent variable as $z$. The decoder is modeling the probabilistic generative processes of $x$ given the continuous representation $z$ through the likelihood $p_\theta(x|z)$ and the prior over the latent variables $p(z)$, where $\theta$ denotes the parameters. The encoder approximates the posterior $p_\theta(z|x) \propto p_\theta(x|z)p(z)$ with a model $q_\psi(z|x)$ parametrized by $\psi$. The decoder and encoder are learned simultaneously by maximizing the evidence lower bound (ELBO) of the marginal likelihood, *i.e.*,

$$\mathcal{L}(X;\theta,\psi) := \sum_{x\in X} \mathbb{E}_{q(z|x)} \left[\log p_\theta(x|z)p(z) - \log q_\psi(z|x)\right] \leq \sum_{x\in X} \log \int p_\theta(x|z)p(z)dz, \quad (1)$$

where $X$ denotes the training datasets containing the observations.

### 2.2 CONTEXT FREE GRAMMAR AND ATTRIBUTE GRAMMAR

**Context free grammar** A context free grammar (CFG) is defined as $G = \langle \mathcal{V}, \Sigma, \mathcal{R}, s \rangle$, where symbols are divided into $\mathcal{V}$, the set of non-terminal symbols, $\Sigma$, the set of terminal symbols and $s \in \mathcal{V}$, the start symbol. Here $\mathcal{R}$ is the set of production rules. Each production rule $r \in \mathcal{R}$ is denoted as $r = \alpha \rightarrow \beta$ for $\alpha \in \mathcal{V}$ is a nonterminal symbol, and $\beta = u_1 u_2 \ldots u_{|\beta|} \in (\mathcal{V} \bigcup \Sigma)^*$ is a sequence of terminal and/or nonterminal symbols.

**Attribute grammar** To enrich the CFG with "semantic meaning", Knuth (1968) formalizes attribute grammar that introduces attributes and rules to CFG. An attribute is an attachment to the corresponding nonterminal symbol in CFG, written in the format $\langle v \rangle.a$ for $v \in \mathcal{V}$. There can be two types of attributes assigned to non-terminals in $G$: the *inherited* attributes and the *synthesized* attributes. An inherited attribute depends on the attributes from its parent and siblings, while a synthesized attribute is computed based on the attributes of its children. Formally, for a production $u_0 \rightarrow u_1 u_2 \ldots u_{|\beta|}$, we denote $I(u_i)$ and $S(u_i)$ be the sets of *inherited* and *synthesized* attributes of $u_i$ for $i \in \{0, \ldots, |\beta|\}$, respectively.

#### 2.2.1 A MOTIVATIONAL EXAMPLE

We here exemplify how the above defined attribute grammar enriches CFG with non-context-free semantics. We use the following toy grammar, a subset of SMILES that generates either a chain or a cycle with three carbons:

**Production**                                         **Semantic Rule**

$\langle s \rangle \quad \rightarrow \langle atom \rangle_1$ 'C' $\langle atom \rangle_2$      $\langle s \rangle.\texttt{matched} \leftarrow \langle atom \rangle_1.\texttt{set} \bigcap \langle atom \rangle_2.\texttt{set},$
$\langle s \rangle.\texttt{ok} \leftarrow \langle atom \rangle_1.\texttt{set} = \langle s \rangle.\texttt{matched} = \langle atom \rangle_2.\texttt{set}$

$\langle atom \rangle \rightarrow$ 'C' $|$ 'C' $\langle bond \rangle \langle digit \rangle$      $\langle atom \rangle.\texttt{set} \leftarrow \varnothing \mid \text{concat}(\langle bond \rangle.\texttt{val}, \langle digit \rangle.\texttt{val})$

$\langle bond \rangle \rightarrow$ '$-$' $|$ '$=$' $|$ '#'      $\langle bond \rangle.\texttt{val} \leftarrow$ '$-$' $|$ '$=$' $|$ '#'

$\langle digit \rangle \rightarrow$ '1' $|$ '2' $|$ ... $|$ '9'      $\langle digit \rangle.\texttt{val} \leftarrow$ '1' $|$ '2' ... $|$ '9'

where we show the production rules in CFG with $\rightarrow$ on the left, and the calculation of attributes in attribute grammar with $\leftarrow$ on the left. Here we leverage the attribute grammar to check (with attribute `matched`) whether the ringbonds come in pairs: a ringbond generated at $\langle atom \rangle_1$ should

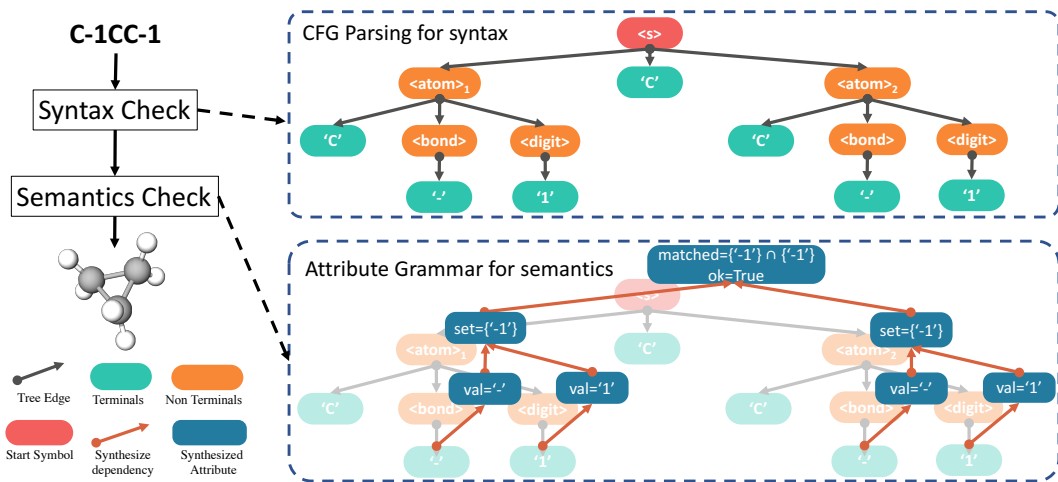

Figure 2: Bottom-up syntax and semantics check in compilers.

match the bond type and bond index that generated at $\langle atom \rangle_2$, also the semantic constraint expressed by $\langle s \rangle$.ok requires that there is no difference between the set attribute of $\langle atom \rangle_1$ and $\langle atom \rangle_2$. Such constraint in SMILES is known as *cross-serial dependencies* (CSD) (Bresnan et al., 1982) which is non-context-free (Shieber, 1985). See Appendix A.3 for more explanations. Figure 2 illustrates the process of performing syntax and semantics check in compilers. Here all the attributes are *synthetic*, *i.e.*, calculated in a bottom-up direction.

So generally, in the semantic correctness checking procedure, one need to perform bottom-up procedures for calculating the attributes *after* the parse tree is generated. However, in the top-down structure generating process, the parse tree is not ready for semantic checking, since the synthesized attributes of each node require information from its children nodes, which are not generated yet. Due to such dilemma, it is nontrivial to use the attribute grammar to guide the top-down generation of the tree-structured data. One straightforward way is using acceptance-rejection sampling scheme, *i.e.*, using the decoder of CVAE or GVAE as a proposal and the semantic checking as the threshold. It is obvious that since the decoder does not include semantic guidance, the proposal distribution may raise semantically invalid candidate frequently, therefore, wasting the computational cost in vain.

## 3    SYNTAX-DIRECTED VARIATIONAL AUTOENCODER

As described in Section 2.2.1, directly using attribute grammar in an offline fashion (*i.e.*, after the generation process finishes) is not efficient to address both syntax and semantics constraints. In this section we describe how to bring forward the attribute grammar online and incorporate it into VAE, such that our VAE addresses both *syntactic* and *semantic* constraints. We name our proposed method Syntax-Directed Variational Autoencoder (SD-VAE).

### 3.1    STOCHASTIC SYNTAX-DIRECTED DECODER

By scrutinizing the tree generation, the major difficulty in incorporating the attributes grammar into the processes is the appearance of the synthesized attributes. For instance, when expanding the start symbol $\langle s \rangle$, none of its children is generated yet. Thus their attributes are also absent at this time, making the $\langle s \rangle$.matched unable to be computed. To enable the on-the-fly computation of the synthesized attributes for semantic validation during tree generation, besides the two types of attributes, we introduce the *stochastic lazy attributes* to enlarge the existing attribute grammar. Such *stochasticity* transforms the corresponding synthesized attribute into inherited constraints in generative procedure; and lazy linking mechanism sets the actual value of the attribute, once all the other dependent attributes are ready. We demonstrate how the decoder with *stochastic lazy attributes* will generate semantic valid output through the same pedagogical example as in Section 2.2.1. Figure 3 visually demonstrates this process.

The tree generation procedure is indeed sampling from the decoder $p_\theta(x|z)$, which can be decomposed into several steps that elaborated below:

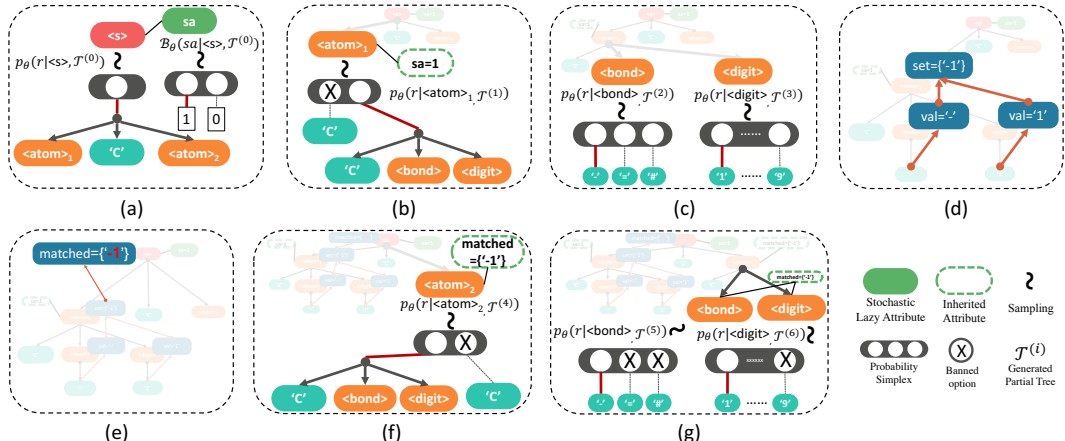

Figure 3: On-the-fly generative process of SD-VAE in order from (a) to (g). Steps: (a) stochastic generation of attribute; (b)(f)(g) constrained sampling with inherited attributes; (c) unconstrained sampling; (d) synthesized attribute calculation on generated subtree. (e) lazy evaluation of the attribute at root node.

---

**Algorithm 1 Decoding with Stochastic Syntax-Directed Decoder**

---

1: **Global variables:** CFG: $G = (\mathcal{V}, \Sigma, \mathcal{R}, s)$, decoder network parameters $\theta$
2: **procedure** GENTREE($node, \mathcal{T}$)
3:     Sample stochastic lazy attribute $node._{sa} \sim \mathcal{B}_\theta(sa|node, \mathcal{T})$      ▷ when introduced on $node$
4:     Sample production rule $r = (\alpha \to \beta) \in \mathcal{R} \sim p_\theta(r|ctx, node, \mathcal{T})$.          ▷ The conditioned variables encodes the semantic constraints in tree generation.
5:     $ctx \leftarrow \text{RNN}(ctx, r)$                                           ▷ update context vector
6:     **for** $i = 1, \ldots, |\beta|$ **do**
7:         $v_i \leftarrow \text{Node}(u_i, node, \{v_j\}_{j=1}^{i-1})$        ▷ node creation with parent and siblings' attributes
8:         GenTree($v_i, \mathcal{T}$)                                ▷ recursive generation of children nodes
9:         Update synthetic and stochastic attributes of $node$ with $v_i$                ▷ Lazy linking
10:     **end for**
11: **end procedure**

---

**i) stochastic predetermination:** in Figure 3(a), we start from the node $\langle s \rangle$ with the synthesized attributes $\langle s \rangle$.matched determining the index and bond type of the ringbond that will be matched at node $\langle s \rangle$. Since we know nothing about the children nodes right now, the only thing we can do is to 'guess' a value. That is to say, we associate a stochastic attribute $\langle s \rangle$.sa $\in \{0, 1\}^{C_a} \sim \prod_{i=1}^{C_a} \mathcal{B}(sa_i|z)$ as a predetermination for the sake of the absence of synthesized attribute $\langle s \rangle$.matched, where $\mathcal{B}(\cdot)$ is the Bernoulli distribution. Here $C_a$ is the maximum cardinality possible [1] for the corresponding attribute $a$. In above example, the $0$ indicates no ringbond and $1$ indicates one ringbond at both $\langle atom \rangle_1$ and $\langle atom \rangle_2$, respectively.

**ii) constraints as inherited attributes:** we pass the $\langle s \rangle$.sa as inherited constraints to the children of node $\langle s \rangle$, *i.e.*, $\langle atom \rangle_1$ and $\langle atom \rangle_2$ to ensure the semantic validation in the tree generation. For example, Figure 3(b) 'sa=1' is passed down to $\langle atom \rangle_1$.

**iii) sampling under constraints:** without loss of generality, we assume $\langle atom \rangle_1$ is generated before $\langle atom \rangle_2$. We then sample the rules from $p_\theta(r|\langle atom \rangle_1, \langle s \rangle, z)$ for expanding $\langle atom \rangle_1$, and so on and so forth to generate the subtree recursively. Since we carefully designed sampling distribution that is conditioning on the stochastic property, the inherited constraints will be eventually satisfied. In the example, due to the $\langle s \rangle$.sa $=$ '1', when expanding $\langle atom \rangle_1$, the sampling distribution $p_\theta(r|\langle atom \rangle_1, \langle s \rangle, z)$ only has positive mass on rule $\langle atom \rangle \to$ 'C' $\langle bond \rangle$ $\langle digit \rangle$.

**iv) lazy linking:** once we complete the generation of the subtree rooted at $\langle atom \rangle_1$, the synthesized attribute $\langle atom \rangle_1$.set is now available. According to the semantic rule for $\langle s \rangle$.matched, we can

---

[1]Note that setting threshold for $C_a$ assumes a *mildly context sensitive grammar* (*e.g.*, limited CSD).

instantiate $\langle s \rangle.\texttt{matched} = \langle atom \rangle_1.\texttt{set} = \{\text{`-1'}\}$. This linking is shown in Figure 3(d)(e). When expanding $\langle atom \rangle_2$, the $\langle s \rangle.\texttt{matched}$ will be passed down as inherited attribute to regulate the generation of $\langle atom \rangle_2$, as is demonstrated in Figure 3(f)(g).

In summary, the general syntax tree $\mathcal{T} \in L(G)$ can be constructed step by step, within the languages $L(G)$ covered by grammar $G$. In the beginning, $\mathcal{T}^{(0)} = root$, where $root._{symbol} = s$ which contains only the start symbol $s$. At step $t$, we will choose an nonterminal node in the *frontier*[2] of partially generated tree $\mathcal{T}^{(t)}$ to expand. The generative process in each step $t = 0, 1, \ldots$ can be described as:

1. Pick node $v^{(t)} \in Fr(\mathcal{T}^{(t)})$ where its attributes needed are either satisfied, or are stochastic attributes that should be sampled first according to Bernoulli distribution $\mathcal{B}(\cdot|v^{(t)}, \mathcal{T}^{(t)})$;

2. Sample rule $r^{(t)} = \alpha^{(t)} \rightarrow \beta^{(t)} \in \mathcal{R}$ according to distribution $p_\theta(r^{(t)}|v^{(t)}, \mathcal{T}^{(t)})$, where $v^{(t)}._{symbol} = \alpha^{(t)}$, and $\beta^{(t)} = u_1^{(t)} u_2^{(t)} \ldots u_{|\beta^{(t)}|}^{(t)}$, *i.e.*, expand the nonterminal with production rules defined in CFG.

3. $\mathcal{T}^{(t+1)} = \mathcal{T}^{(t)} \bigcup \{(v^{(t)}, u_i^{(t)})\}_{i=1}^{|\beta^{(t)}|}$, *i.e.*, grow the tree by attaching $\beta^{(t)}$ to $v^{(t)}$. Now the node $v^{(t)}$ has children represented by symbols in $\beta^{(t)}$.

The above process continues until all the nodes in the frontier of $\mathcal{T}^{(T)}$ are all terminals after $T$ steps. Then, we obtain the algorithm 1 for sampling both syntactic and semantic valid structures.

In fact, in the model training phase, we need to compute the likelihood $p_\theta(x|z)$ given $x$ and $z$. The probability computation procedure is similar to the sampling procedure in the sense that both of them requires tree generation. The only difference is that in the likelihood computation procedure, the tree structure, *i.e.*, the computing path, is fixed since $x$ is given; While in the sampling procedure, it is sampled following the learned model. Specifically, the generative likelihood can be written as:

$$p_\theta(x|z) = \prod_{t=0}^{T} p_\theta(r_t|ctx^{(t)}, node^{(t)}, \mathcal{T}^{(t)})\mathcal{B}_\theta(sa_t|node^{(t)}, \mathcal{T}^{(t)}) \tag{2}$$

where $ctx^{(0)} = z$ and $ctx^{(t)} = \text{RNN}(r_t, ctx^{(t-1)})$. Here RNN can be commonly used LSTM, *etc.*.

## 3.2 STRUCTURE-BASED ENCODER

As we introduced in section 2, the encoder, $q_\psi(z|x)$ approximates the posterior of the latent variable through the model with some parametrized function with parameters $\psi$. Since the structure in the observation $x$ plays an important role, the encoder parametrization should take care of such information. The recently developed deep learning models (Duvenaud et al., 2015; Dai et al., 2016; Lei et al., 2017) provide powerful candidates as encoder. However, to demonstrate the benefits of the proposed syntax-directed decoder in incorporating the attribute grammar for semantic restrictions, we will exploit the same encoder in Kusner et al. (2017) for a fair comparison later.

We provide a brief introduction to the particular encoder model used in Kusner et al. (2017) for a self-contained purpose. Given a program or a SMILES sequence, we obtain the corresponding parse tree using CFG and decompose it into a sequence of productions through a pre-order traversal on the tree. Then, we convert these productions into one-hot indicator vectors, in which each dimension corresponds to one production in the grammar. We will use a deep convolutional neural networks which maps this sequence of one-hot vectors to a continuous vector as the encoder.

## 3.3 MODEL LEARNING

Our learning goal is to maximize the evidence lower bound in Eq 1. Given the encoder, we can then map the structure input into latent space $z$. The variational posterior $q(z|x)$ is parameterized with Gaussian distribution, where the mean and variance are the output of corresponding neural networks. The prior of latent variable $p(z) = \mathcal{N}(0, I)$. Since both the prior and posterior are Gaussian, we use the closed form of KL-divergence that was proposed in Kingma & Welling (2013). In the decoding stage, our goal is to maximize $p_\theta(x|z)$. Using the Equation (2), we can compute the corresponding conditional likelihood. During training, the syntax and semantics constraints required in Algorithm 1

---

[2]Here frontier is the set of all nonterminal leaves in current tree.

can be precomputed. In practice, we observe no significant time penalty measured in wall clock time compared to previous works.

## 4 RELATED WORK

Generative models with discrete structured data have raised increasing interests among researchers in different domains. The classical sequence to sequence model (Sutskever et al., 2014) and its variations have also been applied to molecules (Gómez-Bombarelli et al., 2016). Since the model is quite flexible, it is hard to generate valid structures with limited data, though Dave Janz (2018) shows that an extra validator model could be helpful to some degree. Techniques including data augmentation (Bjerrum, 2017), active learning (Janz et al., 2017) and reinforcement learning (Guimaraes et al., 2017) also been proposed to tackle this issue. However, according to the empirical evaluations from Benhenda (2017), the validity is still not satisfactory. Even when the validity is enforced, the models tend to overfit to simple structures while neglect the diversity.

Since the structured data often comes with formal grammars, it is very helpful to generate its parse tree derived from CFG, instead of generating sequence of tokens directly. The Grammar VAE(Kusner et al., 2017) introduced the CFG constrained decoder for simple math expression and SMILES string generation. The rules are used to mask out invalid syntax such that the generated sequence is always from the language defined by its CFG. Parisotto et al. (2016) uses a RecursiveReverse-Recursive Neural Network (R3NN) to capture global context information while expanding with CFG production rules. Although these works follow the syntax via CFG, the context sensitive information can only be captured using variants of sequence/tree RNNs (Alvarez-Melis & Jaakkola, 2016; Dong & Lapata, 2016; Zhang et al., 2015), which may not be time and sample efficient.

In our work, we capture the semantics with proposed stochastic lazy attributes when generating structured outputs. By addressing the most common semantics to harness the deep networks, it can greatly reshape the output domain of decoder (Hu et al., 2016). As a result, we can also get a better generative model for discrete structures.

## 5 EXPERIMENTS

Code is available at https://github.com/Hanjun-Dai/sdvae.

We show the effectiveness of our proposed SD-VAE with applications in two domains, namely programs and molecules. We compare our method with CVAE (Gómez-Bombarelli et al., 2016) and GVAE (Kusner et al., 2017). CVAE only takes character sequence information, while GVAE utilizes the context-free grammar. To make a fair comparison, we closely follow the experimental protocols that were set up in Kusner et al. (2017). The training details are included in Appendix B.

Our method gets significantly better results than previous works. It yields better reconstruction accuracy and prior validity by large margins, while also having comparative diversity of generated structures. More importantly, the SD-VAE finds better solution in program and molecule regression and optimization tasks. This demonstrates that the continuous latent space obtained by SD-VAE is also smoother and more discriminative.

### 5.1 SETTINGS

Here we first describe our datasets in detail. The programs are represented as a list of statements. Each statement is an atomic arithmetic operation on variables (labeled as $v0, v1, \cdots, v9$) and/or immediate numbers $(1, 2, \ldots, 9)$. Some examples are listed below:

```
v3=sin(v0);v8=exp(2);v9=v3-v8;v5=v0*v9;return:v5
v2=exp(v0);v7=v2*v0;v9=cos(v7);v8=cos(v9);return:v8
```

Here $v0$ is always the input, and the variable specified by `return` (respectively $v5$ and $v8$ in the examples) is the output, therefore it actually represent univariate functions $f : \mathbb{R} \to \mathbb{R}$. Note that a correct program should, besides the context-free grammar specified in Appendix A.1, also respect the semantic constraints. For example, a variable should be defined before being referenced. We

randomly generate $130,000$ programs, where each consisting of 1 to 5 valid statements. Here the maximum number of decoding steps $T = 80$. We hold 2000 programs out for testing and the rest for training and validation.

For molecule experiments, we use the same dataset as in Kusner et al. (2017). It contains $250,000$ SMILES strings, which are extracted from the ZINC database (Gómez-Bombarelli et al., 2016). We use the same split as Kusner et al. (2017), where $5000$ SMILES strings are held out for testing. Regarding the syntax constraints, we use the grammar specified in Appendix A.2, which is also the same as Kusner et al. (2017). Here the maximum number of decoding steps $T = 278$.

For our SD-VAE, we address some of the most common semantics:

**Program semantics** We address the following: *a*) variables should be defined before use, *b*) program must return a variable, *c*) number of statements should be less than 10.

**Molecule semantics** The SMILES semantics we addressed includes: *a*) ringbonds should satisfy cross-serial dependencies, *b*) explicit valence of atoms should not go beyond permitted. For more details about the semantics of SMILES language, please refer to Appendix A.3.

## 5.2 RECONSTRUCTION ACCURACY AND PRIOR VALIDITY

| Methods | Program | | Zinc SMILES | |
|---|---|---|---|---|
| | Reconstruction %* | Valid Prior % | Reconstruction % | Valid Prior % |
| SD-VAE | **96.46 (99.90, 99.12, 90.37)** | **100.00** | **76.2** | **43.5** |
| GVAE | 71.83 (96.30, 77.28, 41.90) | 2.96 | 53.7 | 7.2 |
| CVAE | 13.79 (40.46, 0.87, 0.02) | 0.02 | 44.6 | 0.7 |

Table 1: Reconstructing Accuracy and Prior Validity estimated using Monte Carlo method. Our proposed method (SD-VAE) performance significantly better than existing works.
* We also report the reconstruction % grouped by number of statements (3, 4, 5) in parentheses.

We use the held-out dataset to measure the reconstruction accuracy of VAEs. For prior validity, we first sample the latent representations from prior distribution, and then evaluate how often the model can decode into a valid structure. Since both encoding and decoding are stochastic in VAEs, we follow the Monte Carlo method used in Kusner et al. (2017) to do estimation:

*a*) *reconstruction:* for each of the structured data in the held-out dataset, we encode it 10 times and decoded (for each encoded latent space representation) 25 times, and report the portion of decoded structures that are the same as the input ones; *b*) *validity of prior:* we sample 1000 latent representations $\mathbf{z} \sim \mathcal{N}(O, \mathbf{I})$. For each of them we decode 100 times, and calculate the portion of 100,000 decoded results that corresponds to valid Program or SMILES sequences.

**Program** We show in the left part of Table 1 that our model has near perfect reconstruction rate, and most importantly, a perfect valid decoding program from prior. This huge improvement is due to our model that utilizes the full semantics that previous work ignores, thus in theory guarantees perfect valid prior and in practice enables high reconstruction success rate. For a fair comparison, we run and tune the baselines in $10\%$ of training data and report the best result. In the same place we also report the reconstruction successful rate grouped by number of statements. It is shown that our model keeps high rate even with the size of program growing.

**SMILES** Since the settings are exactly the same, we include CVAE and GVAE results directly from Kusner et al. (2017). We show in the right part of Table 1 that our model produces a much higher rate of successful reconstruction and ratio of valid prior. Figure 8 in Appendix C.2 also demonstrates some decoded molecules from our method. Note that the results we reported have not included the semantics specific to aromaticity into account. If we use an alternative kekulized form of SMILES to train the model, then the valid portion of prior can go up to $97.3\%$.

## 5.3 BAYESIAN OPTIMIZATION

One important application of VAEs is to enable the optimization (*e.g.*, find new structures with better properties) of discrete structures in continuous latent space, and then use decoder to obtain the actual structures. Following the protocol used in Kusner et al. (2017), we use Bayesian Optimization (BO)

to search the programs and molecules with desired properties in latent space. Details about BO settings and parameters can be found in Appendix C.1.

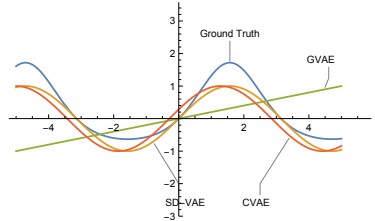

| Method | Program | Score |
|---|---|---|
| CVAE | v7=5+v0;v5=cos(v7);return:v5 | 0.1742 |
| | v2=1-v0;v9=cos(v2);return:v9 | 0.2889 |
| | v5=4+v0;v3=cos(v5);return:v3 | 0.3043 |
| GVAE | v3=1/5;v9=-1;v1=v0*v3;return:v3 | 0.5454 |
| | v2=1/5;v9=-1;v7=v2+v2;return:v7 | 0.5497 |
| | v2=1/5;v5=-v2;v9=v5*v5;return:v9 | 0.5749 |
| SD-VAE | v6=sin(v0);v5=exp(3);v4=v0*v6;return:v6 | **0.1206** |
| | v5=6+v0;v6=sin(v5);return:v6 | **0.1436** |
| | v6=sin(v0);v4=sin(v6);v5=cos(v4);v9=2/v4;return:v4 | **0.1456** |
| Ground Truth | v1=sin(v0);v2=exp(v1);v3=v2-1;return:v3 | — |

Figure 4: On the left are best programs found by each method using Bayesian Optimization. On the right are top 3 closest programs found by each method along with the distance to ground truth (lower distance is better). Both our SD-VAE and CVAE can find similar curves, but our method aligns better with the ground truth. In contrast the GVAE fails this task by reporting trivial programs representing linear functions.

**Finding program** In this application the models are asked to find the program which is most similar to the ground truth program. Here the distance is measured by $\log(1 + \text{MSE})$, where the MSE (Mean Square Error) calculates the discrepancy of program outputs, given the 1000 different inputs v0 sampled evenly in $[-5, 5]$. In Figure 4 we show that our method finds the best program to the ground truth one compared to CVAE and GVAE.

**Molecules** Here we optimize the drug properties of molecules. In this problem, we ask the model to optimize for octanol-water partition coefficients (a.k.a *log P*), an important measurement of drug-likeness of a given molecule. As Gómez-Bombarelli et al. (2016) suggests, for drug-likeness assessment *log P* is penalized by other properties including synthetic accessibility score (Ertl & Schuffenhauer, 2009). In Figure 5 we show the the top-3 best molecules found by each method, where our method found molecules with better scores than previous works. Also one can see the molecule structures found by SD-VAE are richer than baselines, where the latter ones mostly consist of chain structure.

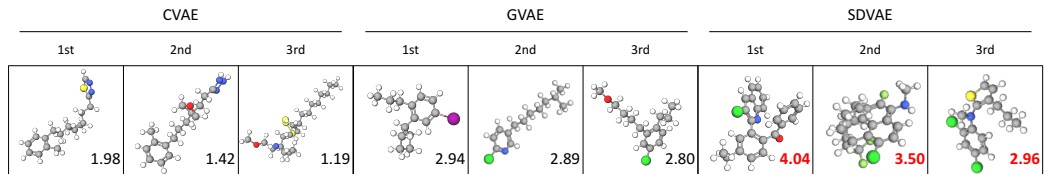

Figure 5: Best top-3 molecules and the corresponding scores found by each method using Bayesian Optimization.

## 5.4 PREDICTIVE PERFORMANCE OF LATENT REPRESENTATION

| Method | Program | | Zinc | |
|---|---|---|---|---|
| | LL | RMSE | LL | RMSE |
| CVAE | -4.943 ± 0.058 | 3.757 ± 0.026 | -1.812 ± 0.004 | 1.504 ± 0.006 |
| GVAE | -4.140 ± 0.038 | 3.378 ± 0.020 | -1.739 ± 0.004 | 1.404 ± 0.006 |
| SD-VAE | **-3.754 ± 0.045** | **3.185 ± 0.025** | **-1.697 ± 0.015** | **1.366 ± 0.023** |

Table 2: Predictive performance using encoded mean latent vector. Test LL and RMSE are reported.

The VAEs also provide a way to do unsupervised feature representation learning Gómez-Bombarelli et al. (2016). In this section, we seek to to know how well our latent space predicts the properties of programs and molecules. After the training of VAEs, we dump the latent vectors of each structured data, and train the sparse Gaussian Process with the target value (namely the error for programs and the drug-likeness for molecules) for regression. We test the performance in the held-out test dataset. In Table 2, we report the result in Log Likelihood (LL) and Regression Mean Square Error (RMSE), which show that our SD-VAE always produces latent space that are more discriminative than both

CVAE and GVAE baselines. This also shows that, with a properly designed decoder, the quality of encoder will also be improved via end-to-end training.

## 5.5 DIVERSITY OF GENERATED MOLECULES

| Similarity Metric | MorganFp | MACCS | PairFp | TopologicalFp |
|---|---|---|---|---|
| GVAE | **0.92 ± 0.10** | **0.83 ± 0.15** | 0.94 ± 0.10 | 0.71 ± 0.14 |
| SD-VAE | **0.92 ± 0.09** | **0.83 ± 0.13** | **0.95 ± 0.08** | **0.75 ± 0.14** |

Table 3: Diversity as statistics from pair-wise distances measured as $1 - s$, where $s$ is one of the similarity metrics. So higher values indicate better diversity. We show $\mathrm{mean} \pm \mathrm{stddev}$ of $\binom{100}{2}$ pairs among 100 molecules. Note that we report results from GVAE and our SD-VAE, because CVAE has very low valid priors, thus completely only failing this evaluation protocol.

Inspired by Benhenda (2017), here we measure the diversity of generated molecules as an assessment of the methods. The intuition is that a good generative model should be able to generate diverse data and avoid mode collapse in the learned space. We conduct this experiment in the SMILES dataset. We first sample 100 points from the prior distribution. For each point, we associate it with a molecule, which is the most frequent occurring valid SMILES decoded (we use 50 decoding attempts since the decoding is stochastic). We then, with one of the several molecular similarity metrics, compute the pairwise similarity and report the mean and standard deviation in Table 3. We see both methods do not have the mode collapse problem, while producing similar diversity scores. It indicates that although our method has more restricted decoding space than baselines, the diversity is not sacrificed. This is because we never rule-out the valid molecules. And a more compact decoding space leads to much higher probability in obtaining valid molecules.

## 5.6 VISUALIZING THE LATENT SPACE

We seek to visualize the latent space as an assessment of how well our generative model is able to produces a coherent and smooth space of program and molecules.

**Program** Following Bowman et al. (2016), we visualize the latent space of program by interpolation between two programs. More specifically, given two programs which are encoded to $p_a$ and $p_b$ respectively in the latent space, we pick 9 evenly interpolated points between them. For each point, we pick the corresponding most decoded structure. In Table 4 we compare our results with previous works. Our SD-VAE can pass though points in the latent space that can be decoded into valid programs without error and with visually more smooth interpolation than previous works. Meanwhile, CVAE makes both syntactic and semantic errors, and GVAE produces only semantic errors (reference of undefined variables), but still in a considerable amount.

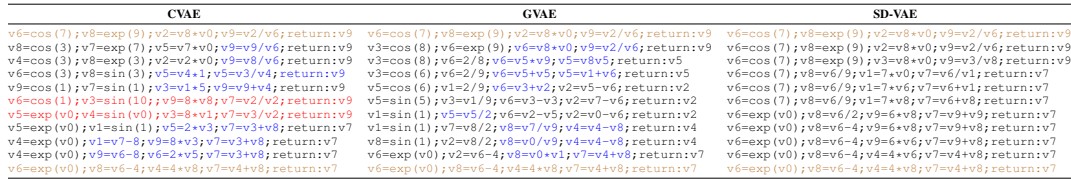

Table 4: Interpolation between two valid programs (the top and bottom ones in brown) where each program occupies a row. Programs in red are with syntax errors. Statements in blue are with semantic errors such as referring to unknown variables. Rows without coloring are correct programs. Observe that when a model passes points in its latent space, our proposed SD-VAE enforces both syntactic and semantic constraints while making visually more smooth interpolation. In contrast, CVAE makes both kinds of mistakes, GVAE avoids syntactic errors but still produces semantic errors, and both methods produce subjectively less smooth interpolations.

**SMILES** For molecules, we visualize the latent space in 2 dimensions. We first embed a random molecule from the dataset into latent space. Then we randomly generate 2 orthogonal unit vectors $A$. To get the latent representation of neighborhood, we interpolate the 2-D grid and project back

to latent space with pseudo inverse of $A$. Finally we show decoded molecules. In Figure 6, we present two of such grid visualizations. Subjectively compared with figures in Kusner et al. (2017), our visualization is characterized by having smooth differences between neighboring molecules, and more complicated decoded structures.

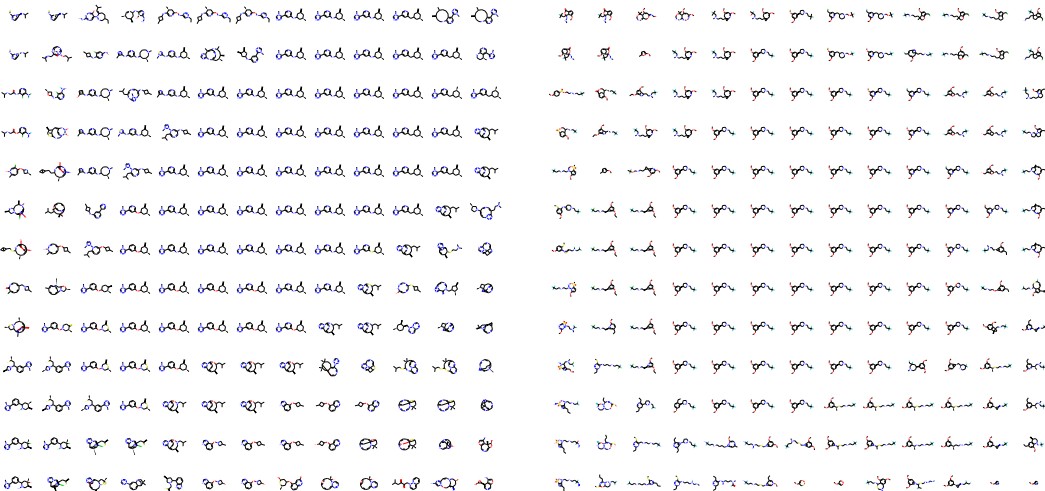

Figure 6: Latent Space visualization. We start from the center molecule and decode the neighborhood latent vectors (neighborhood in projected 2D space).

## 6 CONCLUSION

In this paper we propose a new method to tackle the challenge of addressing both syntax and semantic constraints in generative model for structured data. The newly proposed *stochastic lazy attribute* presents a the systematical conversion from offline syntax and semantic check to online guidance for stochastic generation, and empirically shows consistent and significant improvement over previous models, while requiring similar computational cost as previous model. In the future work, we would like to explore the refinement of formalization on a more theoretical ground, and investigate the application of such formalization on a more diverse set of data modality.

ACKNOWLEDGMENTS

This project was supported in part by NSF IIS-1218749, NIH BIGDATA 1R01GM108341, NSF CAREER IIS-1350983, NSF IIS-1639792 EAGER, NSF CNS-1704701, ONR N00014-15-1-2340, NSF IIS-1546113, DBI-1355990, Intel ISTC, NVIDIA and Amazon AWS.

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

# Appendix

## A    GRAMMAR

### A.1    GRAMMAR FOR PROGRAM SYNTAX

The syntax grammar for program is a generative contest free grammar starting with ⟨*program*⟩.

| ⟨*program*⟩ | → | ⟨*stat list*⟩ |
|---|---|---|
| ⟨*stat list*⟩ | → | ⟨*stat*⟩ ';' ⟨*stat list*⟩ \| ⟨*stat*⟩ |
| ⟨*stat*⟩ | → | ⟨*assign*⟩ \| ⟨*return*⟩ |
| ⟨*assign*⟩ | → | ⟨*lhs*⟩ '=' ⟨*rhs*⟩ |
| ⟨*return*⟩ | → | 'return:' ⟨*lhs*⟩ |
| ⟨*lhs*⟩ | → | ⟨*var*⟩ |
| ⟨*var*⟩ | → | 'v' ⟨*var id*⟩ |
| ⟨*digit*⟩ | → | '1' \| '2' \| '3' \| '4' \| '5' \| '6' \| '7' \| '8' \| '9' |
| ⟨*rhs*⟩ | → | ⟨*expr*⟩ |
| ⟨*expr*⟩ | → | ⟨*unary expr*⟩ \| ⟨*binary expr*⟩ |
| ⟨*unary expr*⟩ | → | ⟨*unary op*⟩ ⟨*operand*⟩ \| ⟨*unary func*⟩ '(' ⟨*operand*⟩ ')' |
| ⟨*binary expr*⟩ | → | ⟨*operand*⟩ ⟨*binary op*⟩ ⟨*operand*⟩ |
| ⟨*unary op*⟩ | → | '+' \| '-' |
| ⟨*unary func*⟩ | → | 'sin' \| 'cos' \| 'exp' |
| ⟨*binary op*⟩ | → | '+' \| '-' \| '*' \| '/' |
| ⟨*operand*⟩ | → | ⟨*var*⟩ \| ⟨*immediate number*⟩ |
| ⟨*immediate number*⟩ | → | ⟨*digit*⟩ '.' ⟨*digit*⟩ |
| ⟨*digit*⟩ | → | '0' \| '1' \| '2' \| '3' \| '4' \| '5' \| '6' \| '7' \| '8' \| '9' |

### A.2    GRAMMAR FOR MOLECULE SYNTAX

Our syntax grammar for molecule is based on OpenSMILES standard, a context free grammar starting with ⟨*s*⟩.

| ⟨*s*⟩ | → | ⟨*atom*⟩ |
|---|---|---|
| ⟨*smiles*⟩ | → | ⟨*chain*⟩ |
| ⟨*atom*⟩ | → | ⟨*bracket atom*⟩ \| ⟨*aliphatic organic*⟩ \| ⟨*aromatic organic*⟩ |
| ⟨*aliphatic organic*⟩ | → | 'B' \| 'C' \| 'N' \| 'O' \| 'S' \| 'P' \| 'F' \| 'I' \| 'Cl' \| 'Br' |
| ⟨*aromatic organic*⟩ | → | 'c' \| 'n' \| 'o' \| 's' |
| ⟨*bracket atom*⟩ | → | '[' ⟨*bracket atom (isotope)*⟩ ']' |
| ⟨*bracket atom (isotope)*⟩ | → | ⟨*isotope*⟩ ⟨*symbol*⟩ ⟨*bracket atom (chiral)*⟩ |
| | \| | ⟨*symbol*⟩ ⟨*bracket atom (chiral)*⟩ |
| | \| | ⟨*isotope*⟩ ⟨*symbol*⟩ \| ⟨*symbol*⟩ |
| ⟨*bracket atom (chiral)*⟩ | → | ⟨*chiral*⟩ ⟨*bracket atom (h count)*⟩ |
| | \| | ⟨*bracket atom (h count)*⟩ |
| | \| | ⟨*chiral*⟩ |
| ⟨*bracket atom (h count)*⟩ | → | ⟨*h count*⟩ ⟨*bracket atom (charge)*⟩ |
| | \| | ⟨*bracket atom (charge)*⟩ |
| | \| | ⟨*h count*⟩ |
| ⟨*bracket atom (charge)*⟩ | → | ⟨*charge*⟩ |
| ⟨*symbol*⟩ | → | ⟨*aliphatic organic*⟩ \| ⟨*aromatic organic*⟩ |
| ⟨*isotope*⟩ | → | ⟨*digit*⟩ \| ⟨*digit*⟩ ⟨*digit*⟩ \| ⟨*digit*⟩ ⟨*digit*⟩ ⟨*digit*⟩ |
| ⟨*digit*⟩ | → | '1' \| '2' \| '3' \| '4' \| '5' \| '6' \| '7' \| '8' |

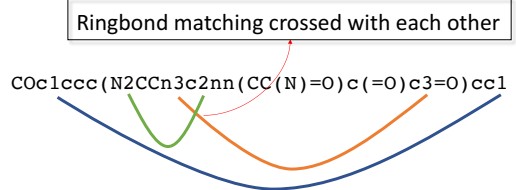

Figure 7: Example of cross-serial dependencies (CSD) that exhibits in SMILES language.

| $\langle chiral \rangle$ | $\rightarrow$ '@' \| '@@' |
|---|---|
| $\langle h\ count \rangle$ | $\rightarrow$ 'H' \| 'H' $\langle digit \rangle$ |
| $\langle charge \rangle$ | $\rightarrow$ '-' \| '-' $\langle digit \rangle$ \| '+' \| '+' $\langle digit \rangle$ |
| $\langle bond \rangle$ | $\rightarrow$ '-' \| '=' \| '#' \| '/' \| '\' |
| $\langle ringbond \rangle$ | $\rightarrow$ $\langle digit \rangle$ |
| $\langle branched\ atom \rangle$ | $\rightarrow$ $\langle atom \rangle$ \| $\langle atom \rangle \langle branches \rangle$ \| $\langle atom \rangle \langle ringbonds \rangle$ \newline \| $\langle atom \rangle \langle ringbonds \rangle \langle branches \rangle$ |
| $\langle ringbonds \rangle$ | $\rightarrow$ $\langle ringbonds \rangle \langle ringbond \rangle$ \| $\langle ringbond \rangle$ |
| $\langle branches \rangle$ | $\rightarrow$ $\langle branches \rangle \langle branch \rangle$ \| $\langle branch \rangle$ |
| $\langle branch \rangle$ | $\rightarrow$ '(' $\langle chain \rangle$ ')' \| '(' $\langle bond \rangle \langle chain \rangle$ ')' |
| $\langle chain \rangle$ | $\rightarrow$ $\langle branched\ atom \rangle$ \| $\langle chain \rangle \langle branched\ atom \rangle$ \newline \| $\langle chain \rangle \langle bond \rangle \langle branched\ atom \rangle$ |

### A.3   EXAMPLES OF SMILES SEMANTICS

Here we provide more explanations of the semantics constraints that contained in SMILES language for molecules.

Specifically, the semantics we addressed here are:

1. **Ringbond matching:** The ringbonds should come in pairs. Each pair of ringbonds has an index and a bond-type associated. What the SMILES semantics requires is exactly the same as the well-known cross-serial dependencies (CSD) in formal language. CSD also appears in some natural languages, such as Dutch and Swiss-German. Another example of CSD is a sequence of multiple different types of parentheses where each separately balanced disregarding the others. See Figure 7 for an illustration.

2. **Explicit valence control:** Intuitively, the semantics requires that each atom cannot have too many bonds associated with it. For example, a normal carbon atom has maximum valence of 4, which means associating a Carbon atom with two triple-bonds will violate the semantics.

### A.4   DEPENDENCY GRAPH INTRODUCED BY ATTRIBUTE GRAMMAR

Suppose there is a production $r = u_0 \rightarrow u_1 u_2 \ldots u_{|\beta|} \in \mathcal{R}$ and an attribute $u_i.a$ we denote the dependency set $D^r(u_i.a) = \{u_j.b | u_j.b$ is required for calculating $u_i.a\}$. The union of all dependency sets $\mathcal{D}_\mathcal{T}^{(att)} = \bigcup_{r \in \mathcal{T}, u_i \in r} D^r(u_i.a)$ induces a dependency graph, where nodes are the attributes and directed edges represents the dependency relationships between those attributes computation. Here $\mathcal{T}$ is an (partial or full) instantiation of the generated syntax tree of grammar $G$. Let $D^r(u_i) = \{u_j | \exists a, b : u_j.b \in D^r(u_i.a)\}$ and $\mathcal{D}_\mathcal{T} = \bigcup_{r \in \mathcal{T}, u_i \in r} D^r(u_i)$, that is, $\mathcal{D}_\mathcal{T}$ is constructed from $\mathcal{D}_\mathcal{T}^{(att)}$ by merging nodes with the same symbol but different attributes, we call $\mathcal{D}_\mathcal{T}^{(att)}$ is noncircular if the corresponding $\mathcal{D}_\mathcal{T}$ is noncircular.

In our paper, we assume the noncircular property of the dependency graph. Such property will be exploited for top-down generation in our decoder.

## B  TRAINING DETAILS

Since our proposed SD-VAE differentiate itself from previous works (CVAE, GVAE) on the formalization of syntax and semantics, we therefore use the same deep neural network model architecture for a fair comparison. In encoder, we use 3-layer one-dimension convolution neural networks (CNNs) followed by a full connected layer, whose output would be fed into two separate affine layers for producing $\mu$ and $\sigma$ respectively as in reparameterization trick; and in decoder we use 3-layer RNNs followed by a affine layer activated by softmax that gives probability for each production rule. In detail, we use 56 dimensions the latent space and the dimension of layers as the same number as in Kusner et al. (2017). As for implementation, we use Kusner et al. (2017)'s open sourced code for baselines, and implement our model in PyTorch framework [3].

In a 10% validation set we tune the following hyper parameters and report the test result from setting with best valid loss. For a fair comparison, all tunings are also conducted in the baselines.

We use `ReconstructLoss` $+ \alpha$`KLDivergence` as the loss function for training. A natural setting is $\alpha = 1$, but Kusner et al. (2017) suggested in their open-sourced implementation[4] that using $\alpha = 1/$`LatentDimension` would leads to better results. We explore both settings.

## C  MORE EXPERIMENT DETAILS

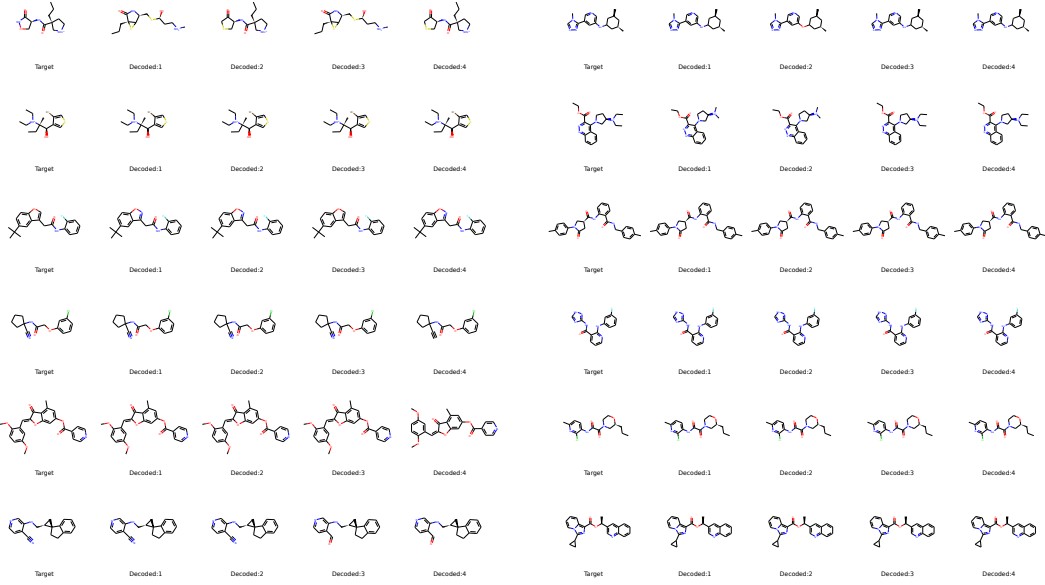

Figure 8: Visualization of reconstruction. The first column in each figure presents the target molecules. We first encode the target molecules, then sample the reconstructed molecules from their encoded posterior.

### C.1  BAYESIAN OPTIMIZATION

The Bayesian optimization is used for searching latent vectors with desired target property. For example, in symbolic program regression, we are interested in finding programs that can fit the given input-output pairs; in drug discovery, we are aiming at finding molecules with maximum drug likeness. To get a fair comparison with baseline algorithms, we follow the settings used in Kusner et al. (2017).

---

[3] `http://pytorch.org/`
[4] `https://github.com/mkusner/grammarVAE/issues/2`

Specifically, we first train the variational autoencoder in an unsupervised way. After obtaining the generative model, we encode all the structures into latent space. Then these vectors and corresponding property values (*i.e.*, estimated errors for program, or drug likeness for molecule) are used to train a sparse Gaussian process with 500 inducing points. This is used later for predicting properties in latent space. Next, 5 iterations of batch Bayesian optimization with the expected improvement (EI) heuristic is used for proposing new latent vectors. In each iteration, 50 latent vectors are proposed. After the proposal, the newly found programs/molecules are then added to the batch for next round of iteration.

During the proposal of latent vectors in each iteration, we perform 100 rounds of decoding and pick the most frequent decoded structures. This helps regulates the decoding due to randomness, as well as increasing the chance for baselines algorithms to propose valid ones.

## C.2 RECONTRUCTION

We visualize some reconstruction results of SMILES in Figure 8. It can be observed that, in most cases the decoder successfully recover the exact origin input. Due to the stochasticity of decoder, it may have some small variations.

