# OpenReview forum: "Syntax-Directed Variational Autoencoder for Structured Data"
_ICLR.cc/2018/Conference — Accept (Poster)_

### Official Review · AnonReviewer2 · 2017-11-27
**Interesting idea but poor presentation**

**Rating:** 3
**Confidence:** 2

**Review:**

The paper presents an approach for improving variational autoencoders for structured data that provide an output that is both syntactically valid and semantically reasonable.  The idea presented seems to have merit , however, I found the presentation lacking. Many sentences are poorly written making the paper hard to read, especially when not familiar with the presented methods. The experimental section could be organized better. I didn't like that two types of experiment are now presented in parallel. Finally, the paper stops abruptly without any final discussion and/or conclusion.

---

> ### Author Response · Authors · 2017-12-07
> **Reply to "Interesting idea but poor presentation"**
>
> We thank you for providing reviews.
>
> We’ll refine the paper to include more introduction about background, and more detailed explanations about our method.
>
> We’ll include final discussion/conclusion section.

---

> > ### Comment · AnonReviewer2 · 2018-01-12
> > **Presentation improved but still lacking**
> >
> > The presentation of the paper has definately improved, but I find the language used in the paper still below the quality needed for publication. There are still way too many grammatical and syntactical errors.

---

### Official Review · AnonReviewer1 · 2017-11-27

**Rating:** 5
**Confidence:** 1

**Review:**

Let me first note that I am not very familiar with the literature on program generation,
molecule design or compiler theory, which this paper draws heavily from, so my review is an educated guess.

This paper proposes to include additional constraints into a VAE which generates discrete sequences,
namely constraints enforcing both semantic and syntactic validity.
This is an extension to the Grammar VAE of Kusner et. al, which includes syntactic constraints but not semantic ones.
These semantic constraints are formalized in the form of an attribute grammar, which is provided in addition to the context-free grammar.
The authors evaluate their methods on two tasks, program generation and molecule generation.

Their method makes use of additional prior knowledge of semantics, which seems task-specific and limits the generality of their model.
They report that their method outperforms the Character VAE (CVAE) and Grammar VAE (GVAE) of Kusner et. al.
However, it isn't clear whether the comparison is appropriate: the authors report in the appendix that they use the kekulised version of the Zinc dataset of Kusner et. al, whereas Kusner et. al do not make any mention of this.
The baselines they compare against for CVAE and GVAE in Table 1 are taken directly from Kusner et. al though.
Can the authors clarify whether the different methods they compare in Table 1 are all run on the same dataset format?

Typos:
- Page 5: "while in sampling procedure" -> "while in the sampling procedure"
- Page 6: "a deep convolution neural networks" -> "a deep convolutional neural network"
- Page 6: "KL-divergence that proposed in" -> "KL-divergence that was proposed in"
- Page 6: "since in training time" -> "since at training time"
- Page 6: "can effectively computed" -> "can effectively be computed"
- Page 7: "reset for training" -> "rest for training"

---

> ### Author Response · Authors · 2017-12-07
> **Reply to "Review"**
>
> Thanks for your effort in providing this detailed and useful review!
>
> We present our clarification in the following:
>
> >> Use of data and comparison with baselines:
>
> We would first note that the anonymous accusation was set to “17 Nov 2017 (modified: 28 Nov 2017), readers: ICLR 2018 Conference Reviewers and Higher”. That’s why it was not visible to us until Nov 28, i.e., the original review release date. This gives us no chance to clarify anything before the review deadline. We have replied to it actively since Nov 28.
> **Note the thread is invisible to us again since Dec 2. **
>
> 1) We have experimented both kekulization and non-kekulization for baselines, and have reported the best they can get in all experiments. For example, in Table 2 the GVAE baseline results are improved compared to what was reported in GVAE paper.
>
> 2) The anonymous commenter is using different kekulization (RDKIT, rather than our used Marvin), different baseline implementation (custom implementation, rather than the public one in GVAE’s paper) and possibly different evaluation code (since there is no corresponding evaluation online). For a reproducible comparision, we released our implementation, data, pretrained model and evaluation code at:  https://github.com/anonymous-author-80ee48b2f87/cvae-baseline
>
> 3) To make further clarification, we ran our method on the vanilla (non-kekulised) data. Our performance is actually boosted (76.2% vs 72.8% reported in the paper).
> The details of results from these experiments above can be seen in our public reply titled “We released baseline CVAE code, data and evaluation code for clarification” and “Our reconstruction performance without kekulization on Zinc dataset”.
>
> In either setting still, our method outperforms all baselines on reconstruction. We are sorry that this may have led to some confusions. To avoid further possible misunderstandings, we have extensively rerun all experiments involving ZINC dataset. Though differences are observed, the conclusion in each experiment remains the same. For example, our reconstruction performance is boosted (76.2% vs 72.8%). Since we didn’t address aromaticity semantics by the paper submission deadline, the valid prior fraction drops to 43.5%, but it is still much higher than baselines (7.2% GVAE, 0.7% CVAE). Please find the updated paper for more details.
>
> >> prior knowledge and limitations
>
> We are targeting on domains where strict syntax and semantics are required. For example, the syntax and semantics are needed to compile a program, or to parse a molecule structure. So such prior knowledge comes naturally with the application. Our contribution is to incorporate such existing syntax and semantics in those compilers, into an on-the-fly generation process of structures.
>
> In general, when numerous amount of data is available, a general seq2seq model would be enough. However, obtaining the useful drug molecules is expensive, and thus data is quite limited. Using knowledges like syntax (e.g., in GVAE paper), or semantics (like in our paper) will greatly reduce the amount of data needed to obtain a good model.
>
> In our paper, we only addressed 2-3 semantic constraints, where the improvement is significant. Similarly, in “Harnessing Deep Neural Networks with Logic Rules (Hu et.al, ACL 16)”, incorporating several intuitive rules can greatly improve the performance of sentiment analysis, NER, etc. So we believe that, incorporating the knowledge with powerful deep learning achieves a good trade-off between human efforts and model performance.
>
> >> Typos and other writing issue:
>
> We thank you very much for your careful reading and pointing out the typos and writing issues in our manuscript! We have incorporated your suggested changes in the current revision, and are keeping conducting further detailed proofreading to fix as much as possible the writing issues in the future revisions.

---

### Official Review · AnonReviewer3 · 2017-11-27
**Strong paper presents state-of-the-art results**

**Rating:** 7
**Confidence:** 3

**Review:**

NOTE:

Would the authors kindly respond to the comment below regarding Kekulisation of the Zinc dataset? Fair comparison of the data is a serious concern. I have listed this review as a good for publication due to the novelty of ideas presented, but the accusation of misrepresentation below is a serious one and I would like to know the author's response.

*Overview*

This paper presents a method of generating both syntactically and semantically valid data from a variational autoencoder model using ideas inspired by compiler semantic checking. Instead of verifying the semantic correctness offline of a particular discrete structure, the authors propose “stochastic lazy attributes”, which amounts to loading semantic constraints into a CFG and using a tailored latent-space decoder algorithm that guarantees both syntactic semantic valid. Using Bayesian Optimization, search over this space can yield decodings with targeted properties.

Many of the ideas presented are novel. The results presented are state-of-the art. As noted in the paper, the generation of syntactically and semantically valid data is still an open problem. This paper presents an interesting and valuable solution, and as such constitutes a large advance in this nascent area of machine learning.

*Remarks on methodology*

By initializing a decoding by “guessing” a value, the decoder will focus on high-probability starting regions of the space of possible structures. It is not clear to me immediately how this will affect the output distribution. Since this process on average begins at high-probability region and makes further decoding decisions from that starting point, the output distribution may be biased since it is the output of cuts through high-probability regions of the possible outputs space. Does this sacrifice exploration for exploitation in some quantifiable way? Some exploration of this issue or commentary would be valuable.

*Nitpicks*

I found the notion of stochastic predetermination somewhat opaque, and section 3 in general introduces much terminology, like lazy linking, that was new to me coming from a machine learning background. In my opinion, this section could benefit from a little more expansion and conceptual definition.

The first 3 sections of the paper are very clearly written, but the remainder has many typos and grammatical errors (often word omission). The draft could use a few more passes before publication.

---

> ### Author Response · Authors · 2017-12-07
> **Reply to "Strong paper presents state-of-the-art results"**
>
> Thanks for your effort in providing this detailed and constructive review!
> We present our clarification in the following:
>
> >>NOTE:
>
> We would first note that the anonymous accusation was set to “17 Nov 2017 (modified: 28 Nov 2017), readers: ICLR 2018 Conference Reviewers and Higher”. That’s why it was not visible to us until Nov 28, i.e., the original review release date. This gives us no chance to clarify anything before the review deadline. We have replied to it actively since Nov 28.
> **Note the thread is invisible to us again since Dec 2. **
>
> To summarize our clarification:
>
> >> Use of data
>
> 1) We have experimented both kekulization and non-kekulization for baselines, and have reported the best they can get in all experiments. For example, in Table 2 the GVAE baseline results are improved compared to what was reported in GVAE paper.
>
> 2) The anonymous commenter is using different kekulization (RDKIT, rather than our used Marvin), different baseline implementation (custom implementation, rather than the public one in GVAE’s paper) and possibly different evaluation code (since there is no corresponding evaluation online). For a reproducible comparision, we released our implementation, data, pretrained model and evaluation code at:  https://github.com/anonymous-author-80ee48b2f87/cvae-baseline
>
> 3) To make further clarification, we ran our method on the vanilla (non-kekulised) data. Our performance is actually boosted (76.2% vs 72.8% reported in the paper).
> The details of results from these experiments above can be seen in our public reply titled “We released baseline CVAE code, data and evaluation code for clarification” and “Our reconstruction performance without kekulization on Zinc dataset”.
>
> In either setting still, our method outperforms all baselines on reconstruction. We are sorry that this may have led to some confusions. To avoid further possible misunderstandings, we have extensively rerun all experiments involving ZINC dataset. Though differences are observed, the conclusion in each experiment remains the same. For example, our reconstruction performance is boosted (76.2% vs 72.8%). Since we didn’t address aromaticity semantics by the paper submission deadline, the valid prior fraction drops to 43.5%, but it is still much higher than baselines (7.2% GVAE, 0.7% CVAE). Please find the updated paper for more details.
>
> >>sacrifice of exploration
>
> CVAE, GVAE and our SD-VAE are all factorizing the joint probability of entire program / SMILES text in some way. CVAE factorizes in char level, GVAE in Context Free Grammar (CFG) tree, while ours factorizes both CFG and non-context free semantics. Since every method is factorizing the entire space, each structure in this space should have the possibility (despite its magnitude) of being sampled.
>
> Bias is not always a bad thing. Some bias will help the model quickly concentrate to the correct mode. Definitely, different methods will bias the distribution in a different way. For example, CVAE is biased towards the beginning of the sequence. GVAE is biased by several initial non-terminals.
>
> Our experiments on diversity of generated molecules (table 3) demonstrate that, both GVAE and our method can generate quite diverse molecules. So we think both methods don’t have noticeable mode collapse problem on this dataset.
>
> >> writings:
>
> Thanks for the suggestions. We are adding more effort in explaining our algorithm and improve writing in revisions. We have revised our experiments sections for clarifying the most important issue, and will keep improving the writing.
>
> To briefly answer the “lazy linking”: We don’t sample the actual value of the attribute at the first encounter; Instead, later when the actual content is generated, we use bottom-up calculation to fill the value. For example, when generating ringbond attribute, we only sample its existence. The ringbond information (bond index and bond type) are filled later.
>
> As a side note, this idea comes from “lazy evaluation” in compiler theory where a value is not calculated until it is needed.

---

### Author Response · Authors · 2017-11-30
**We released baseline CVAE code, data and evaluation code for clarification**

To address the anonymous commenter’s concerns on the CVAE baseline, the initial release of CVAE’s code (training code based on GVAE’s authors’code), with two versions of kekule data and vanilla data and the reconstruction evaluation script,  are available at

https://github.com/anonymous-author-80ee48b2f87/cvae-baseline

where we also uploaded our trained CVAE, together with pretrained model obtained from GVAE’s authors.

Here we briefly summarize the current results:
(1) - CVAE, vanilla setting, pretrained model : 44.854%
(2) - CVAE, vanilla setting, our retraining: 43.218%
(3) - CVAE, Marvin Suite kekulised **tried for all methods in our paper**: 11.6%
(4) - CVAE, rdkit kekulised (provided by anonymous commenter, never been tried in our paper): 38.17%

We reported the best form of SMILES for CVAE in our paper. If you believe there’s any issue, please let us know asap and we are happy to investigate.

Finally, we thank all the anonymous comments about the paper. If you have any concerns about the paper, please make the comments public while you specifying readers. Making such comments to reviewers only will not allow us to address the possible misunderstandings, or improve the paper timely when we make possible mistakes.

---

### Author Response · Authors · 2017-12-01
**Our reconstruction performance without kekulization on Zinc dataset**

To further clarify the reconstruction accuracy, we here report performance (our model and baselines) without using the kekulization transformation on Zinc dataset, in supplement to numbers using kekulization already reported in our manuscript. We include baseline results from GVAE paper for direct comparison.

SD-VAE (ours): 76.2%; GVAE: 53.7%; CVAE: 44.6%

Compare to what reported for SD-VAE with kekulization in current revision (72.8%), our performance is slightly boosted without kekulization. This shows that kekulization itself doesn’t have positive impact for reconstruction in our method. Our conclusion that the reconstruction accuracy of our SD-VAE is much better than all baselines still holds.

Nevertheless, to avoid possible misunderstanding, we’ll refine the experiment section by including more experiments, once the open review system allows.

---

### Author Response · Authors · 2017-12-07
**Paper revision 1**

To avoid further possible misunderstandings we have update our paper, in which we have extensively revised all experiments involving ZINC dataset. This addresses concerns on use of ZINC data and comparison with previous methods.

The conclusion in each experiment **remains the same** though some differences are observed. Examples of differences are as following: Our reconstruction performance is boosted (76.2% vs 72.8%); And since we didn’t address semantics specific to aromaticity by the paper submission deadline, the valid prior fraction drops to 43.5%, but it is still much higher than baselines (7.2% GVAE, 0.7% CVAE).

Please find the updated paper for more details.

---

### Author Response · Authors · 2018-01-05
**Paper revision 2**

In addition to our revision 1, in which we extensively revised all experiments involving ZINC dataset, we have made an updated revision 2 which mostly addresses the writing and presentation issues. Besides the refinement of wording and typos, this version includes the following modification:

1) We added Figure 2, where we explicitly show how the modern compiler works through the example of two-stage check (i.e., CFG parsing and Attribute Grammar check). Section 2 is now augmented with more detailed explanations of background knowledge.

2) We added Figure 3, which shows the proposed syntax-directed decoder step by step through an example. Through the examples we put more effort in explaining key concepts in our method, such as ‘inherited constraints’ and ‘lazy linking’.

3) Experiment section is revised with more details included.

4) We added a conclusion section as suggested by the reviewer.

---

### Decision · Program_Chairs · 2018-01-29
**ICLR 2018 Conference Acceptance Decision**

**Decision:**

Accept (Poster)

**Comment:**

This paper presents a more complex version of the grammar-VAE, which can be used to generate structured discrete objects for which a grammar is known, by adding a second 'attribute grammar', inspired by Knuth.

Overall, the idea is a bit incremental, but the space is wide open and I think that structured encoder/decoders is an important direction.  The experiments seem to have been done carefully (with some help from the reviewers) and the results are convincing.